# Comparative Estimation of the Cytotoxic Activity of Different Parts of *Cynara scolymus* L.: Crude Extracts versus Green Synthesized Silver Nanoparticles with Apoptotic Investigation

**DOI:** 10.3390/pharmaceutics14102185

**Published:** 2022-10-13

**Authors:** Amgad I. M. Khedr, Abdelaziz F. S. Farrag, Ali M. Nasr, Shady A. Swidan, Mohamed S. Nafie, Maged S. Abdel-Kader, Marwa S. Goda, Jihan M. Badr, Reda F. A. Abdelhameed

**Affiliations:** 1Department of Pharmacognosy, Faculty of Pharmacy, Port Said University, Port Said 42526, Egypt; 2Department of Pharmaceutics, Faculty of Pharmacy, Port Said University, Port Said 42526, Egypt; 3Department of Pharmaceutics and Industrial Pharmacy, Faculty of Pharmacy, Galala University, New Galala 43713, Egypt; 4Department of Pharmaceutics and Pharmaceutical Technology, Faculty of Pharmacy, The British University in Egypt, El-Sherouk City, Cairo 11837, Egypt; 5The Centre for Drug Research and Development (CDRD), Faculty of Pharmacy, The British University in Egypt, El-Sherouk City, Cairo 11837, Egypt; 6Department of Chemistry, Faculty of Science, Suez Canal University, Ismailia 41522, Egypt; 7Department of Pharmacognosy, College of Pharmacy, Prince Sattam Bin Abdulaziz University, Al-Kharj 11942, Saudi Arabia; 8Department of Pharmacognosy, Faculty of Pharmacy, Alexandria University, Alexandria 21215, Egypt; 9Department of Pharmacognosy, Faculty of Pharmacy, Suez Canal University, Ismailia 41522, Egypt; 10Department of Pharmacognosy, Faculty of Pharmacy, Galala University, New Galala 43713, Egypt

**Keywords:** *Cynara scolymus* L., biomedical implementations, cytotoxic, green synthesis, silver nanoparticles

## Abstract

Different parts of *Cynara scolymus* L. and their green synthesized eco-friendly silver nanoparticles (AgNPs) were screened for their cytotoxicity and apoptotic activity. Results showed that flower extract AgNPs exhibited more potent cytotoxicity compared to the normal form against PC-3 and A549 cell lines with IC_50_ values of 2.47 μg/mL and 1.35 μg/mL, respectively. The results were compared to doxorubicin (IC_50_ = 5.13 and 6.19 μg/mL, respectively). For apoptosis-induction, AgNPs prepared from the flower extract induced cell death by apoptosis by 41.34-fold change and induced necrotic cell death by 10.2-fold. Additionally, they induced total prostate apoptotic cell death by a 16.18-fold change, and it slightly induced necrotic cell death by 2.7-fold. Hence, green synthesized flower extract AgNPs exhibited cytotoxicity in A549 and PC-3 through apoptosis-induction in both cells. Consequently, synthesized AgNPs were further tested for apoptosis and increased gene and protein expression of pro-apoptotic markers while decreasing expression of anti-apoptotic genes. As a result, this formula may serve as a promising source for anti-cancer candidates. Finally, liquid chromatography combined with electrospray mass spectrometry (LC-ESI-MS) analysis was assessed to identify the common bioactive metabolites in crude extracts of stem, flower, and bract.

## 1. Introduction

*Cynara scolymus L*. is a perennial plant belonging to the family Compositae, and it grows in a big head from edible buds with multiple triangular scales of 8–15 cm in diameter. It is cultivated in the Mediterranean region for the fresh, immature flower head. As a vegetable, it is consumed as fresh, tinned, or frozen [1,2]. The plant is of a great nutritional value as it accumulates a considerable amount of folic acid, essential amino acids, and fatty acids, of which the most plentiful are the n-6 linoleic and palmitic acids [3]. *C. Scolymus* extracts have long been used in folk medicine for their great choleretic and hepatoprotective properties in addition to their efficiency to ameliorate bacterial infection, arteriosclerosis, and diabetes symptoms [4,5,6]. Chemically, *Cynara scolymus L.* compiles a precious secondary metabolite to which the pharmacological activities are attributed. These compounds include phenolic acids, namely chlorogenic acid, cynarin, and caffeic acid; bitter sesquiterpene lactones (for example, cynaropicrin and grosheimin); flavonoids (luteolin, luteolin-7-O-rutinoside, luteolin-7-O-β-gluco-pyranoside, apigenin-7-O-rutinoside); and cynarasaponins and inulin [7,8]. As a result, the plant has the potency to be exploited in phytopharmaceutical applications. Recent advances in drug manufacture have led to evolution of various herbal drug delivery systems [9]. The revival of interest in the nanotechnology arena has led to new developments in terms of nanoparticle biosynthesis. The biosynthesis approach is preferable over physiochemical synthesis methods owing to its biocompatibility and cost-effectiveness and also because it is ecofriendly. The biological technique is one of the most widely used methods in the green synthesis of metallic nanoparticles, specifically silver nanoparticles (AgNPs). More recently, it has been proved that the plant-mediated green synthesis of AgNPs has grown into a novel branch of nanotechnology. Green synthesis is a novel approach which overcomes the limitations of classical physiochemical methods by utilizing a wide variety of natural herbs which are biocompatible and nontoxic [10]. Generally, AgNPs are nanoparticles of silver atoms having a size distribution range between 1 and 100 nm and possessing unique electrical, magnetic, and optical characteristics having broad potential applications [6]. Although different noble metals have been employed for a variety of uses, AgNPs have been concentrated on for possible uses in the detection and treatment of cancer. There are two main requirements for green synthesis of AgNPs. The first factor is the dissolving of the silver metal ion and secondly a reducing biological agent should be included in the synthesis process. In most cases, reducing agents in the cells work as a stabilizing and capping agent and so there is no requirement to add external capping stabilizing agents [11]. Because the physicochemical parameters of AgNPs have a substantial impact on their biological behavior in vivo, precise characterization should be performed after synthesis. It is vital to evaluate the manufactured nanoparticles prior to administration in order to use the maximum performance of each nanoparticle safely and without any biological issues. AgNPs’ biological and cytotoxic activity is primarily influenced by a number of variables, such as the size distribution, particle shape, rate of dissolution, and kind of reducing agents utilized in the synthesis of the AgNPs. In addition, they affect cellular uptake and internalization, biological distribution, and biological barrier penetration. Therefore, accurate design and synthesis of AgNPs that are uniform in size, morphology, and functionality are essential for wide variety of biomedical applications. In the current study, we aim to investigate methods to potentiate the cytotoxic effect of different *Cynara scolymus L.* part extracts through green synthesis of AgNPs. The biological and cytotoxic activity of AgNPs is based on several factors as particle morphology, size distribution, dissolution rate, and the type of reducing agents utilized in the AgNPs synthesis [12,13]. Moreover, they influence cellular uptake, biological distribution, and biological barrier penetration [14]. Consequently, accurate design and synthesis of AgNPs which possess uniformity in size, morphology, and functionality are fundamental for biomedical implementations [15].

## 2. Materials and Methods

### 2.1. Plant Extracts

The plant was collected from El Behera Governorate, which has the largest cultivated area of artichokes in Egypt, during December 2020. Then, it was taxonomically identified at Faculty of Science, Suez Canal University, and a voucher specimen was kept at the herbarium of Pharmacognosy Department, Faculty of Pharmacy, Suez Canal University under registration code (CS-2020). Three different parts were detached from the plant to be investigated separately: bracts, flowers, and stems. Each part was air dried by placing it on a shallow tray lined with a layer of paper towels in a dry place with good air circulation for 7 to 10 days, and then they were finely ground using an electric grinder. This step was followed by cold maceration (I Kg of each part) at room temperature for 1 week using methanol. The extraction process was repeated three times to ensure complete extraction [16]. The three extracts were concentrated under reduced pressure using rotary evaporator and stored in the refrigerator.

### 2.2. Determination of the Total Phenolic Content in the Different Plant Extracts

The total phenolics in the three different extracts (bracts, flowers and stem) of *Cynara scolymus* L. were assessed spectrophotometrically using Folin–Ciocalteu method, as previously mentioned [17]. UV absorbance was measured at λ 630 nm using a Milton Roy, Spectronic 1201 (Houston, TX, USA). Gallic acid was utilized as a standard. The results were reported in terms of gallic acid equivalents (mg GAE/g dry extract).

### 2.3. Green Synthesis and Preparation of Silver Nanoparticles

The biogenic synthesis of AgNPs in the presence of the total extract of bracts, flowers, and stems were prepared using a modified method of that reported by Kim et al. [18]. and Ashour et al. [19]. Initially, 10 mg of the extract was dissolved in 1 mL ethanol, then added to 10 mL of 10 mM AgNO_3_. A few drops of 1 M NaOH were added, and the mixture was agitated for 1 h at 400 rpm at 60 °C in the dark. All prepared nanoparticles were purified by centrifugation at 15,000 rpm for 1 h at 4 °C. The AgNPs were re-dispersed in double-distilled water and sonicated for 30 s in sonicating water bath, then centrifuged under the same previous conditions. The washing procedures using double-distilled water were repeated three times.

### 2.4. Characterization of Silver Nanoparticles

#### 2.4.1. UV-Vis Absorbance Spectroscopy

The reduction of Ag+ ions was confirmed by measuring the UV-vis spectrum using a double-beam spectrophotometer (V630, Jasco, Tokyo, Japan). The spectrum was recorded throughout a range of 300–600 nm.

#### 2.4.2. Size Analysis and Surface Charge Determination

Average particle size (expressed as Z-average), zeta potential (ZP) and polydispersity index (PDI) were accurately measured by photon correlation spectroscopy (PCS) using Malvern Zetasizer (Nano ZS, Malvern Instruments Ltd., Malvern, UK). Each sample was diluted 20 times with distilled water before analysis. All measurements were performed at ambient temperature (25 °C) and in triplicates. Finally, the mean and standard deviation values were accurately calculated [20].

#### 2.4.3. Transmission Electron Microscopy (TEM)

Transmission electron microscopy (TEM) was carried out to examine the size and surface morphology of the synthesized AgNPs. The sample preparations were further diluted 50 times with double-distilled water. Then the diluted samples were negatively stained with phosphotungstic acid and dried on carbon-coated copper grids. The thin film formed was air-dried at room temperature and observed using transmission electron microscope (JTEM model 1010, JEOL^®^, Tokyo, Japan) with an accelerating voltage of about 100 kV [21].

### 2.5. Biological Activity

#### 2.5.1. MTT Assay

Cancer cell lines A549 and PC-3 were obtained from the National Cancer Institute in Cairo, Egypt, cultured in “RPMI-1640/DMEM media L-Glutamine (Lonza Verviers SPRL, Belgium, cat#12-604F). The cells were cultured in 10% fetal bovine serum (FBS, Sigma-Aldrich, Burlington, MA, USA) and 1% penicillin/streptomycin (Lonza, Verviers, Belgium)”. Cells were seeded in triplicate on a 96-well plate at a density of 5 × 10^4^ cells, and then treated with the extracts at concentrations of (0.1, 1, 10, and 100 g/mL) for 72 h. Cell viability was assessed using MTT solution (Promega, Madison, WI, USA) [22,23].

#### 2.5.2. Annexin V/PI Staining for Apoptosis/Necrosis Assessment

Treatment with flower extract AgNPs (IC_50_ = 1.52 µM, 48 h) was applied to A549 cells that had been cultured in 6-well culture plates (3–5 × 10^5^ cells/well) overnight. Following collection of cells and medium supernatants, the cells were suspended in 100 L of Annexin-binding buffer solution “25 mM CaCl_2_, 1.4 M NaCl, and 0.1 M Hepes/NaOH, pH 7.4” and incubation with “Annexin V-FITC solution (1:100) and propidium iodide (PI) at a concentration equal to 10 µg/mL in the dark for 30 min.” Afterwards, the Cytoflex FACS machine was used to collect the labelled cells, and the cytExpert software was utilized to evaluate the results [24,25,26].

#### 2.5.3. Gene Expression Analysis (RT-PCR) for the Selected Genes

Gene expression of P53, Bax, and Caspases-3,8,9 was identified as pro-apoptotic genes, while Bcl-2 was identified as an anti-apoptotic gene, and their gene expression was assessed using RT-PCR to investigate the apoptotic pathway; their sequences in forward and reverse direction are shown in Table 1. MCF-7 cells were treated with flower extract AgNPs (IC_50_ = 1.52 µM, 48 h) following the usual procedures, and the RT-PCR reaction was carried out [27,28].

#### 2.5.4. Protein Expression Using Western Blotting

Furthermore, untreated, and treated A549 cancer cells were treated with flower extract AgNPs (IC_50_ = 1.52 µM, 48 h) were tested for protein expression using Western blotting assay for further validation of the apoptotic pathway at the protein expression level. Cells were washed in PBS and lysed in boiling sample buffer before being electrophoresed on a sodium dodecyl sulphate polyacrylamide gel (SDS-PAGE). After boiling the lysates for 5 min in lamellae buffer, the proteins were separated using SDS-PAGE and transferred to an Immobilon membrane (Millipore, Merck KGaA, Darmstadt, Germany). After blocking in 5% nonfat milk for 1 h, the membranes were probed overnight at 4 °C with primary antibodies against “P53, Bax, Bcl-2, caspase-3, and caspase-9”. After conjugating appropriate secondary antibodies, immunoblots were densitometrically analyzed to quantify the amounts of tested proteins [29].

### 2.6. Liquid Chromatography–Electrospray Ionization Mass Spevtrometry (LC-ESI-MS) Analysis

The LC-ESI-MS analysis was performed using a UPLC instrument equipped with a reversed-phase C-18 column (ACQUITY UPLC BEH C18 column, 1.7 μm particle size, 2.1 × 50 mm column). Mobile phase elution was conducted at a flow rate of 0.2 mL/min using a gradient mobile phase (A: H_2_O, B: acetonitrile (Merck, Darmstadt, Germany), both acidified with 0.1% formic acid (Merck, Darmstadt, Germany)). The UPLC compartment was equipped with an electrospray ESI source (electrospray voltage, 3.0 kV; sheath gas, nitrogen; capillary temperature, 440 °C) in negative ionization mode. The ESI-MS range was set at *m/z* 100 and 1000, with starting collision-induced dissociation energy of 30 eV, and MassLynx 4.1 software was used to process the spectra. The compounds were identified by comparison of their data, particularly accurate masses, with those reported in the literature [30].

## 3. Results

### 3.1. Total Phenolic Content in the Different Plant Extracts

The total phenolic contents of the three different extracts of *Cynara scolymus L*. were assessed separately using the Folin–Ciocalteu colorimetric method. The results were presented as gallic acid equivalents and found as 74.29 ± 3.85, 60.94 ± 3.28 and 26.59 ± 1.37 (mg/gm) for the flower, bract, and stem extracts, respectively, where the flower extract revealed the highest content compared to the bracts and the stem extracts.

### 3.2. UV-Vis Absorbance Spectroscopy

All prepared AgNPs formulations showed brown color due to the characteristic surface-plasmon resonance absorption band in the range of 400–500 nm. Figure 1. It is known that metal nanoparticles have a surface plasmon resonance absorption in the UV–Visible region. The band of surface plasmon occurs due to the coherent existence of free electrons in the conduction band due to the small particle size [31,32]. This confirms the reduction of Ag+ ion to colloidal Ag.

### 3.3. Size Analysis and Surface Charge Determination

Measurement of mean particle size by DLS technique gives a comprehensive picture of the particle size of the whole sample as well as the homogeneity of size distribution throughout the sample. That is why it was performed in the current study, although it is reported that DLS measurement is mostly used for AgNPs synthesized from bio-polymers not from the plant extracts and microorganisms [11]. Results given in Table 2 show that particle size values of the prepared AgNPs ranged between 23.6 ± 1.08 of the LL extract AgNPs and 27.2 ± 0.91 nm for the AA extract AgNPs. From the obtained data, it is clear that the three prepared formulations show small difference in the mean particle size. This can be an indication to the consistency of the method of preparation and the absence of effect on different parts of the plant on the obtained AgNPs size. It is well proven that the particle size, surface charge, and NPs shape have a significant impact on pharmacokinetics, cell internalization, tissue distribution, cellular uptake, and clearance. Additionally, physiological processes such as hepatic uptake and tissue diffusion, tissue extravasation, and renal excretion mainly depend on average particle size [33]. Additionally, controlling the particle size, charge, and surface chemistry of the nanocarrier enables to avoid various limitations of conventional treatments such as administration of higher doses, low bioavailability, and the poor chemical stability of the administered drug [34]. It is also reported that the NPs size is one of the most important factors used to predict the circulation time inside living tissues. The interaction of macrophages with targets nanoparticles is dependent on particle size, as shown by Doshi et al. [35]. It was reported in the literature that this small range of particle sizes showed higher cytotoxicity compared to larger particles. Xu and colleagues studied the effect of particle size of the AgNPs on different glioma cell lines. They found that the AgNPs of size range (20–50 nm) was found to be cytotoxic to glioma cells compared to 100 nm particles [36]. Liu et al. investigated the effect of AgNPs of different particle sizes on four different human cell lines. They found that the smaller nanoparticles enter cells more easily than larger ones, which may be the cause of the higher toxic effects [37].

On the other hand, PDI measures size distribution in the formula and ranges usually from 0 to 1. Low PDI values ranged from 0.123 ± 0.006 to 0.283 ± 0.020 show a narrow size distribution and encourage long-term nano dispersion stability; however, values greater than 0.5 suggest that there is no uniform size distribution [38]. Lower values of PDI are desirable for a lower variation in AgNPs formulation. The PDI of all prepared AgNPs formulae show uniform size distribution and perfect homogeneity; see Table 2.

Zeta potential is the overall charge acquired by the particles. It is important to obtain precise judgments about nanoparticles dispersions stability. The colloidal dispersions are considered highly stable when the ZP value is around 30 mV or greater due to the presence of electrostatic repulsion between particles [25]. A high value of zeta potential will ensure system stability, which helps the nanocarrier to resist aggregation. When the zeta potential is very low, attractive forces exceed repulsion and the dispersion will be unstable. So, nanoparticles with higher zeta potentials are electrically stabilized [39]. In this investigation, the results obtained for zeta potential were in the range between −27.2 ± 1.417 and −34.2 ± 0.66 mV, as illustrated in Table 2. The ZP results clearly indicate that all prepared AgNPs have sufficient charges that would prevent their agglomeration and are considered highly stable.

### 3.4. Transmission Electron Microscopy (TEM)

The surface morphology, size and shape of the synthesized AgNPs was illustrated by TEM. The TEM image Figure 2 revealed that AgNPs are spherical in shape, fairly monodispersed, and effectively dispersed without agglomeration. The inset of Figure 2 shows the particle size distribution of AgNPs derived from TEM data. The barely perceptible variation in PS measurement between the two techniques (TEM vs. DLS) results can be illustrated by the difference in the measurements techniques. TEM measurement is a number-based particle size measuring method however DLS is an intensity-based method.

### 3.5. Cytotoxic Activity

Figure 3 shows the results of an MTT assay testing the cytotoxicity of crude extract samples from the brackets, flowers, and stems of Cynara scolymus L. against prostate (PC-3) and lung (A549) cancer cell lines. Based on the cytotoxicity data shown in Table 3, crude extracts of brackets, flowers, and stems revealed low to moderate cytotoxic activity against A549 and PC-3 cells (IC_50_ = 36.57 to 165.3 g/mL). When compared to their non-nano counterparts, cytotoxicity results were significantly enhanced when nano formulations were used. Flower AgNPs displayed strong cytotoxicity against PC-3 and A549 cell lines, with IC_50_ values of 2.47 g/mL and 1.35 g/mL, respectively, compared to doxorubicin and AgNPs with IC_50_ values of 5.13 and 3.75 μg/mL against PC-3 cells, and with IC_50_ values of 6.19 and 22 μg/mL against A549 cells, respectively. Additionally, AgNPs of the bracts showed promising cytotoxicity with IC_50_ values of 14.29 and 16.4 μg/mL, while AgNPs of the stems showed weak cytotoxicity. These findings demonstrated that the floral extract’s AgNPs possessed significant cytotoxicity, justifying additional investigation into the mechanism of action in A549 and PC-3 cells.

### 3.6. Apoptosis-Induction Activity

#### 3.6.1. Annexin V/PI Staining

Cytotoxic activity of flower extract AgNPs against A549 and PC-3 cells was investigated for its mechanism for apoptosis-induction using Annexin V/PI staining. As seen in Figure 4, flower extract AgNPs induced total lung apoptotic cell death by 27.7% compared to 0.67% in the untreated control cells. These findings showed that this extract induced cell death by apoptosis by a 41.34-fold change, and it induced necrotic cell death by 10.2-fold. Additionally, it induced total prostate apoptotic cell death by 17.8% compared to 1.1% in the untreated control cells. These findings showed that this extract induced cell death by apoptosis by a 16.18-fold change, and it slightly induced necrotic cell death by 2.7-fold. Hence, flower extract AgNPs exhibited cytotoxicity in A549 and PC-3 through apoptosis-induction in both cells.

#### 3.6.2. Gene Expression Analysis using RT-PCR

Gene expression analysis for apoptosis-related genes in untreated and treated A549 cells was performed to examine apoptosis-induction by flower extract AgNPs treatment. As seen in Figure 5, flower extract AgNPs upregulated P53 gene by 7.9-fold, Bax gene by 6.98-fold, and caspases 3, 8, 9 by 10.76, 2.84, and 8.47-fold, respectively, while it downregulated the Bcl-2 gene by 0.19-fold, this treatment-induced apoptosis in A549 cells was consistent with expected behavior [24,40] of proving apoptosis-induction.

#### 3.6.3. Protein Expression Using Western Blotting

Further confirmation of apoptosis-induction in A549 cells using Western blotting was employed to measure the levels of p53, Bax, caspase-3 and 9, and Bcl-2 proteins. P53, Bax, and caspase-3 and 9 proteins were found to be upregulated in the study results. Both thick and thin bands and their relative quantification data from Figure 6 show that flower extract AgNPs treatment resulted in a decrease in Bcl-2 protein expression. These findings are in line with the upregulation of pro-apoptotic genes and the downregulation of anti-apoptotic genes observed during RT-PCR.

Previous studies revealed that there is a statistically significant relationship between total polyphenol consumption and the risk of developing cancer [41]. Due to the fact that the phenolic sets in polyphenols can receive electron with formation of steady phenoxy radical, polyphenols disturb or disorganize series of oxidation reactions in constituents of cells, and this is responsible for their ability to prevent various degenerative diseases, including different cancer types [42,43]. Moreover, it was declared that phenolic acids exert a direct anti-proliferative action [44]. It was also declared that the flowers of *Cynara scolymus L.* accumulate several phenolic acids including quinic, chlorogenic, gallic, ferulic, cinnamic vanillic, 1,5-Dicaffeoylquinic, and 1,3-di-O-caffeoylquinic acid (cynarin) [45,46]. The flower of the plant also contains a number of flavonoids, mainly luteolin and apigenin (both aglycones) and glycosides [47,48]. Based on the data on our present study, the flower extract of *Cynara scolymus L.* exhibited the highest content of phenolic constituents exactly 74.29 ± 3.85 (mg/gm) compared to 60.94 ± 3.28 and 26.59 ± 1.37 (mg/gm) for the bract and stem extracts, respectively. This can explain the relatively higher cytotoxic activity of the flower extract compared to the other two extracts. Since the aim of our work is to optimize the cytotoxic activity and to discover the most efficient way to make use of this natural product, we selected the flower extract to be formulated as AgNPs. As mentioned in the discussion, this formula exhibited more potent cytotoxic effects, and these results make it recommended as an optimistic drug lead.

### 3.7. UPLC-ESI-MS Analysis for Identifiction of Bioactive Metabolites

The three parts (flower, bract, and stem) of *Cynara Scolymus* were analyzed using UPLC-ESI-MS, resulting in a noticeable difference in their chemical composition, as shown in Figure 7 and Table 4.

## 4. Discussion

The crude extract of the floral portion of *C. scolymus* L. has the greatest levels of total phenolics and total flavonoids as reported in the literature by Palermo et al. [58]. Previous studies reported that the flower is especially rich in phenolic compounds such as quercetin and 1,3-di-O-Caffeoylquinic acid (cynarin) [45,49,54]. Bracts contain Apigenin-7-O-malonyl glucoside, hesperidin, and Cynaropicrin, as reported in [41]. Flower and bracts contain Luteo-lin-7-O-glucoside (cynaroside), 3-O-Caffeoylquinic acid (Neo Chlorogenic acid). 5-O-Caffeoylquinic acid (Chlorogenic acid), Luteolin 7-O-malonylglucoside, 1,5-di-O-Caffeoylquinic acid, 3,5-Di-O-caffeoylquinic Acid, also reported in [41,53,54] Chlorogenic Acid, Luteolin, luteolin-7-*O*-rutinoside and Apigenin are existing in the three parts with considerable difference in concentration as reported in [41,45,56], and this finding agrees with our study. These polyphenols exhibited cytotoxic effects and apoptosis-induction activity that help in the treatment of prostate and lung cancer. It is now obvious why the crude extract of the *C. scolymus* L. flower portion showed a stronger cytotoxic activity than those of the bracts or stem. Previous studies as those by Mileo, Di Venere et al. conducted studies on the edible part (head) of fresh artichoke using breast cancer cell line MDA-MB231 and found that high dosages of polyphenolic extracts of artichoke extracts can initiate an apoptotic pathway and halt tumor development [59]. Additionally, Pulito, Mori et al. tested the edible parts (receptacles with inner and intermediate bracts) and leaves using cell cytotoxicity assay on human carcinoma cells MSTO-211H, MPP-89, and NCI-H28 mesothelioma cell lines and found that *Cynara scolymus* affects malignant pleural mesothelioma by promoting apoptosis and restraining invasion [60]. Through the use of intrinsic and extrinsic signalling pathways, luteolin and luteolin-7-O-glucoside (cynaroside) can slow or stop the growth of cancer cells both in vitro and in vivo by protecting against carcinogenic stimuli, inhibiting tumor cell proliferation, inducing cell cycle arrest, and inducing apoptosis [61,62]. Through several investigations, quercetin (QU) and quercetin glycoside (rutin), two polyphenolic flavonoids, stand out among the natural products. The biological characteristics of rutin (RU) and its aglycone (QU) include anti-oxidant, anti-inflammatory, and anti-carcinogenic actions [63]. Chlorogenic acid (CGA), also known as 5-O-caffeoylquinic acid, is polyphenol act as curative compounds against oxidative stress, which is a key player in the pathophysiology of many diseases, including cancer [64]. Moreover, 1,5-di-O-Caffeoylquinic acid, 3,5-Di-O-caffeoylquinic acid, and other phenolic compounds possess potent anti-RSV activity, anti-oxidant and anti-bacterial activity [65,66,67]. The primary caffeoylquinic acid derivative, cynarin, is present in the leaves and heads of artichokes. It may have cytotoxic, choleretic, anti-oxidative, hepatoprotective, anti-atherosclerotic, and anti-oxidative properties [68]. Extensive studies have demonstrated the significant anti-oxidant, anti-inflammatory, and anti-carcinogenic capabilities of apigenin. [69].

In this study, there are three mechanisms for evaluation of in vitro cytotoxic activity as Annexin V/PI staining, Gene expression analysis using RT-PCR [24,40] or protein expression using Western blotting. These methods provide simplicity, cost-management, and rapid technique. The comparison between the extract from *Cynara scolymus*’ various sections and its nano form is the study’s main objective. The potential for cytotoxicity increases with decreasing IC_50_ values for the crude extracts under investigation. The biological findings showed that, in comparison to doxorubicin as a reference drug, the silver nano particles of polyphenolic fraction of flower extract (flower extract AgNPs) exhibited potent cytotoxicity against PC-3 and A549 cell lines. Using flow cytometry and gene expression tests to investigate apoptosis-induction, flower extract AgNPs demonstrated a greater ratio of apoptosis in PC-3 cells compared to A549 cells. Our results agreed with their reported anti-cancer activities Phuong Thuy et al. and Shallan et al., [70,71] either through apoptosis-induction or anti-oxidant activation. Erdogan et al. [72] previously created silver nanoparticles using *Cynara scolymus* leaf extracts and shown their anti-cancer potential by inducing apoptosis. Finally, these positive results suggest more in vivo research on the effectiveness of silver nanoparticles of the total phenolic fraction of *C. scolymus* L. flower extract against prostate cancer. The estimate of the polyphenolic fraction of floral extract’s pharmacokinetics is another interesting area of research.

## 5. Conclusions

*Cynara scolymus* L. flower extract AgNPs exhibited potent cytotoxicity compared to the normal form against PC-3 and A549 cell lines with IC_50_ values of 2.47 μg/mL and 1.35 μg/mL, respectively. It exhibited apoptosis activity in both A549 and PC-3 cells by 41.34 and 16.18-fold-changes. Additionally, apoptosis activity was further proved through upregulation of pro-apoptotic markers of P53, Bax, caspases-3, 8, 9, and downregulation of anti-apoptotic marker Bcl-2 in both gene and protein expression levels. So, this formula may serve as a promising source for anti-cancer candidates.

## Figures and Tables

**Figure 1 pharmaceutics-14-02185-f001:**
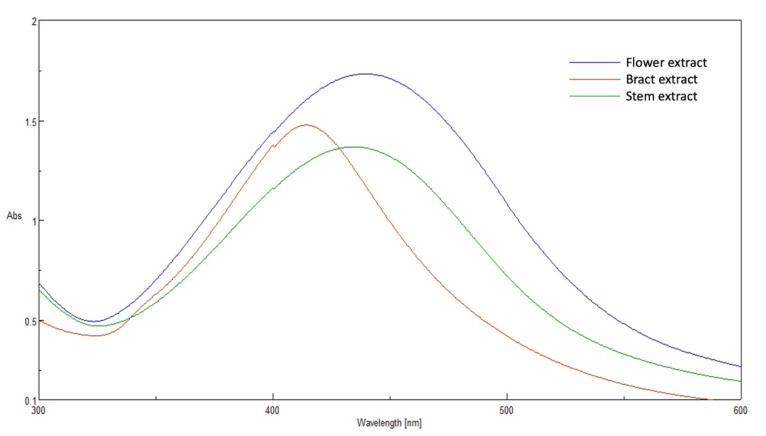
The UV-visible spectrum of prepared AgNPs using extracts of different parts of *Cynara scolymus*.

**Figure 2 pharmaceutics-14-02185-f002:**
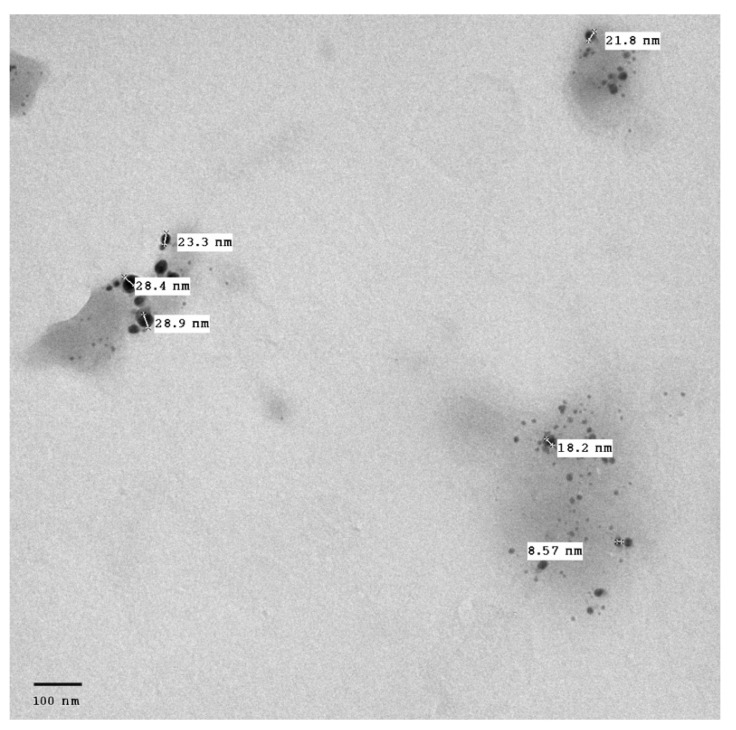
TEM micrograph of *Cynara scolymus* flower extract AgNPs. Mag. 80,000×.

**Figure 3 pharmaceutics-14-02185-f003:**
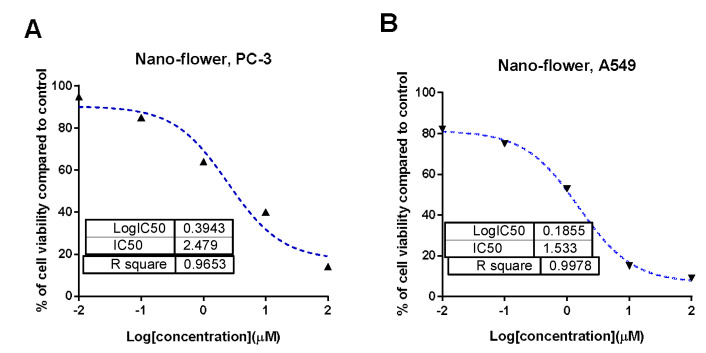
Percentage of cell viability vs. log [con. µM], R square ≈ 1 using the GraphPad prism software. (**A**) cytotoxicity of flower extract AgNPs against PC-3 cells. (**B**) cytotoxicity of flower extract AgNPs against A549 cells.

**Figure 4 pharmaceutics-14-02185-f004:**
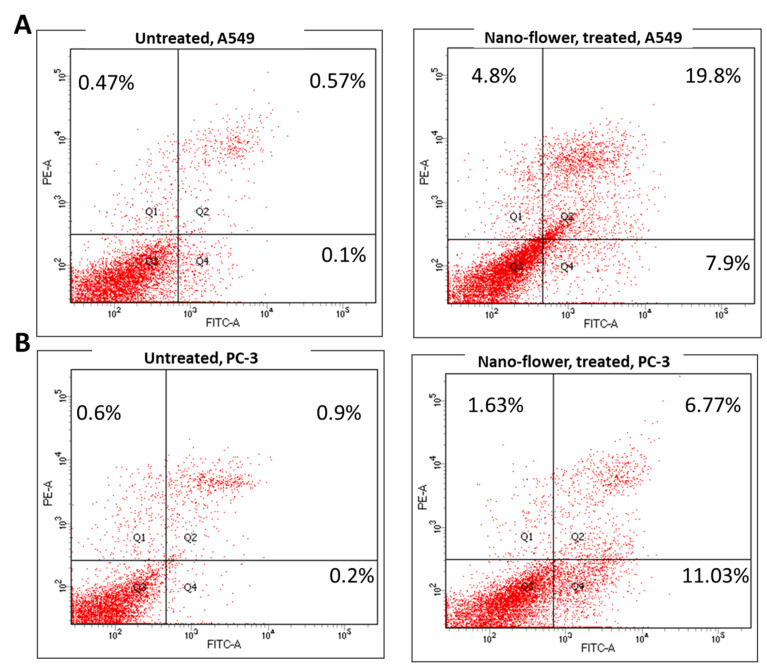
Cytograms for apoptosis–necrosis assessment using flow cytometry. (**A**) Annexin V/PI staining of untreated and treated A549 cancer cells with flower extract AgNPs (IC_50_ = 1.52 µM, 48 h). (**B**) Annexin V/PI staining of untreated and treated A549 cancer cells with flower extract AgNPs (IC_50_ = 2.47 µM, 48 h). Q1: necrosis, Q2: late apoptosis, Q4: early apoptosis (lower panel).

**Figure 5 pharmaceutics-14-02185-f005:**
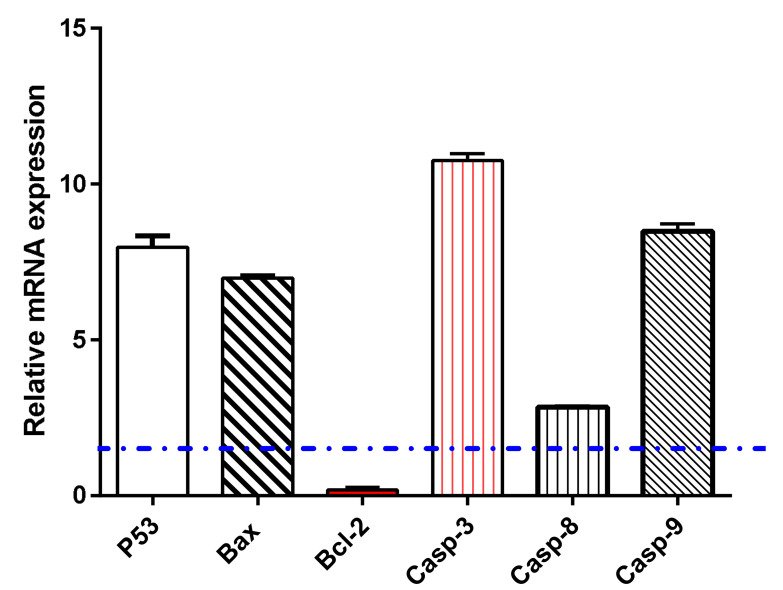
Evaluation of flower extract AgNPs on A549 cell gene expression in both control and treated samples (IC_50_ = 1.52 µM, 48 h). β-actin was used as a house-keeping gene. Dashed line represents for the fold change of untreated control. Fold of change is calculated by 2^ ^-ΔΔCT^, where ΔΔCT is the difference between mean values of genes CT values in the treated and control groups.

**Figure 6 pharmaceutics-14-02185-f006:**
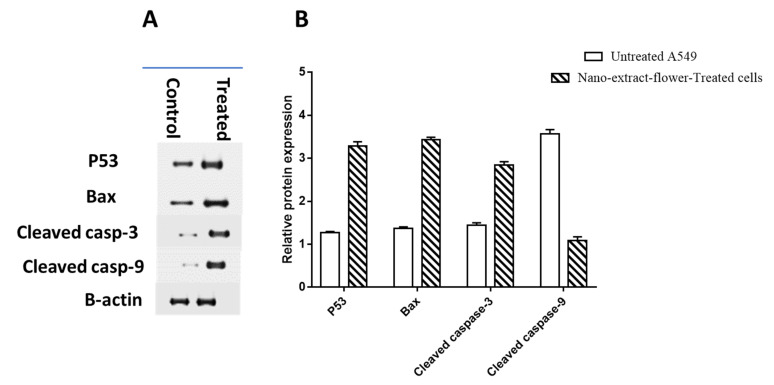
Protein expression level in the untreated and treated A549 cells with flower extract AgNPs (IC_50_ = 1.52 µM, 48 h). (**A**) Gel images and (**B**) quantification of band width. Data normalization was made relative to β-actin.

**Figure 7 pharmaceutics-14-02185-f007:**
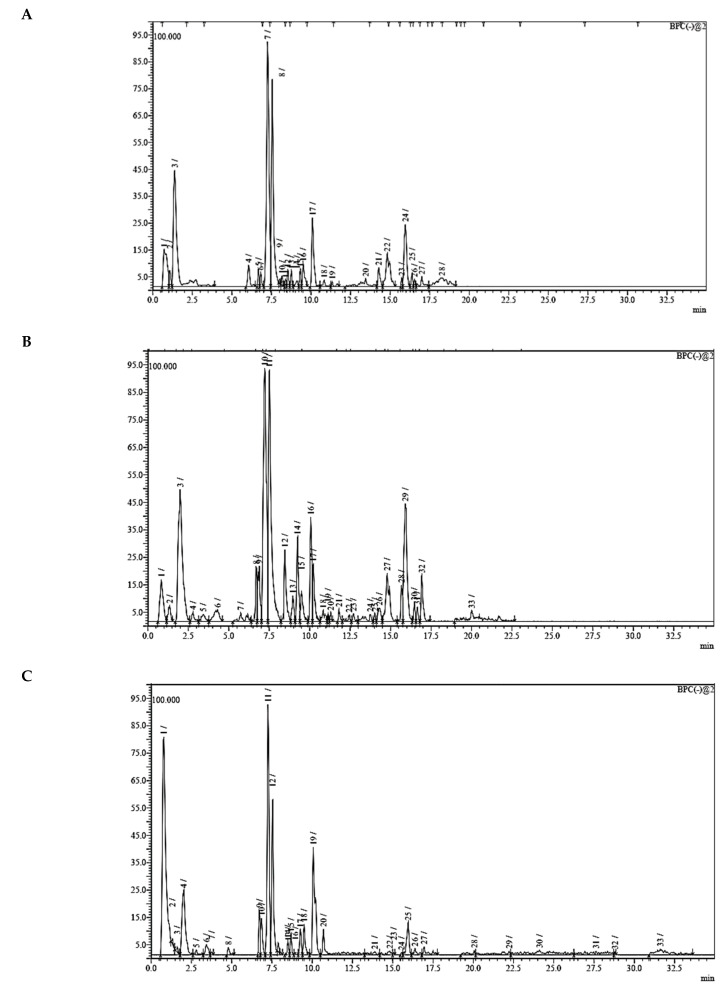
UPLC-ESI-MS chromatograms of crude extracts of bract (**A**), flower (**B**), and stem (**C**) of *C. scolymus* L.

**Table 1 pharmaceutics-14-02185-t001:** Sequences of forward and reverse primers.

Gene	Forward	Reverse
P53	5′-CCCCTCCTGGCCCCTGTCATCTTC-3′	5′-GCAGCGCCTCACAACCTCCGTCAT-3′
Bax	5′-GTTTCATCCAGGATCGAGCAG-3′	5′-CATCTTCTTCCAGATGGTGA-3′
CASP-3	5′-TGGCCCTGAAATACGAAGTC-3′	5′-GGCAGTAGTCGACTCTGAAG-3′
CASP-8	5′-AATGTTGGAGGAAAGCAAT-3′	5′-CATAGTCGTTGATTATCTTCAGC-3′
CASP-9	5′-CGAACTAACAGGCAAGCAGC-3′	5′- ACCTCACCAAATCCTCCAGAAC-3′
Bcl-2	5′-CCTGTGGATGACTGAGTACC-3′	5′-GAGACAGCCAGGAGAAATCA-3′
β-actin	5′-GTGACATCCACACCCAGAGG-3′	5′-ACAGGATGTCAAAACTGCCC-3′

**Table 2 pharmaceutics-14-02185-t002:** Particle size (PS), polydispersity index (PDI), and zeta potential (ZP) of the synthesized AgNPs.

Formula	PS (nm)	PDI	ZP (mV)
Flower extract AgNPs	26.57 ± 0.431	0.204 ± 0.027	−29.9 ± 0.854
Bract extract AgNPs	23.60 ± 1.082	0.123 ± 0.006	−34.2 ± 0.666
Stem extract AgNPs	27.24 ± 0.912	0.283 ± 0.020	−27.2 ± 1.417

**Table 3 pharmaceutics-14-02185-t003:** Summarized IC_50_ values for different extracts with AgNPs of *Cynara scolymus* L.

	Sample	Working Concentration	IC_50_ [μg/mL] *	
			PC-3	A549
1	Extract “bracts”	0.1, 1, 10, 50, 100 μg/mL	68.3 ± 2.1	86.4 ± 3.1
2	Extract “flowers”		45.36 ± 2.6	36.57 ± 1.56
3	Extract “stems”		165.3 ± 4.96	136.1 ± 4.9
4	Bract extract AgNPs		14.29 ± 1.23	16.4 ± 0.72
5	Flower extract AgNPs		2.47 ± 0.24	1.53 ± 0.34
6	Stem extract AgNPs		83.4 ± 2.19	61.2 ± 2.65
7	AgNPs		3.75 ± 0.32	22.0 ± 1.12
8	Doxorubicin		5.13 ± 0.64	6.19 ± 0.58

* IC_50_ were calculated by non-linear regression curve fir using GraphPad prism.

**Table 4 pharmaceutics-14-02185-t004:** LC-MS metabolomic analysis of methanolic crude extracts of bract, flower, and stem of *C. scolymus* L.

Ret. Time	*m/z*	Adduct	MolecularFormula	Deduced Compound	References
**Bract**
1.38	353	[M-H]^−^	C_16_H_18_O_9_	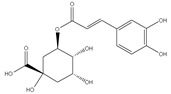 chlorogenic acid	[41,42,43,44,45,46,47,48,49]
3.42	472	[M-2H]^−^	C_23_H_22_O_11_	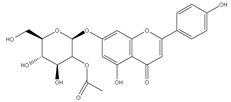 apigenin-7-*O*-acetyl-glucoside	[41]
5.74	533	[M-H]^−^	C_24_H_22_O_14_	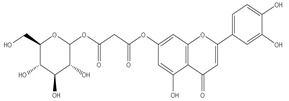 Luteolin 7-*O*-malonylglucoside	[41,47]
6.69	593	[M-2H]^−^	C_27_H_30_O_15_	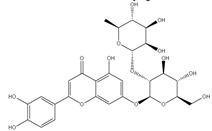 luteolin-7-*O*-rutinoside	[41,49]
7.26	515	[M-H]^−^	C_25_H_24_O_12_	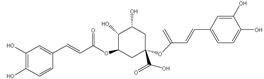 cynarin	[41,49]
8.01	425	[M-H]^−^	C_21_H_30_O_9_	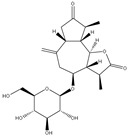 cynarascoloside c	[41,50]
8.53	285	[M-H]^−^	C_15_H_10_O_6_	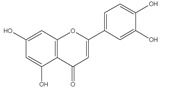 luteolin	[41,51]
9.33	269	[M-H]^−^	C_15_H_10_O_5_	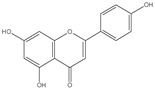 apigenin	[41,51]
10.83	329	[M-H]^−^	C_18_H_34_O_5_	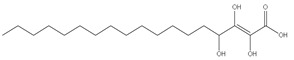 trihydroxyoctadecenoic acid	[41,52]
13.45	345	[M-H]^−^	C_19_H_22_O_6_	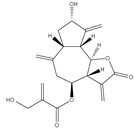 cynaropicrin	[45,53]
14.79	293	[M-H]^−^	C_18_H_30_O_3_	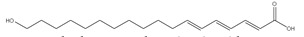 hydroxy-octadecatrienoic acid	[41]
**Flower**
1.95	353	[M-H]^−^	C_16_H_18_O_9_	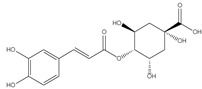 cryptochlorogenic acid	[53]
2.79	179	[M-H]^−^	C_9_H_8_O_4_	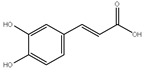 caffeic acid	[45]
6.70	593	[M-H]^−^	C_27_H_30_O_15_	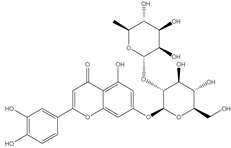 luteolin-7-*O*-rutinoside	[45]
6.88	925	[M-H]^−^	C_47_H_74_O_18_	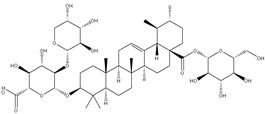 Cynarasaponin A	[45]
6.88	925	[M-H]^−^	C_47_H_74_O_18_	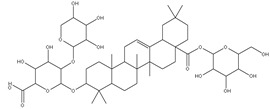 cynarasaponin H	[45]
7.23	515	[M-H]^−^	C_25_H_24_O_12_	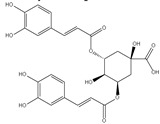 3,5-Di-*O*-caffeoylquinic acid	[54]
7.51	515	[M-H]^−^	C_25_H_24_O_12_	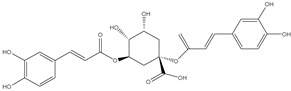 cynarin	[45,54]
8.46	285	[M-H]^−^	C_15_H_10_O_6_	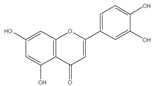 luteolin	[44,45,55]
8.97	779	[M-H]^−^	C_41_H_64_O_14_	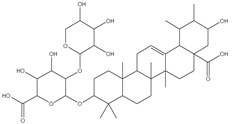 cynarasaponin F	[45]
9.25	269	[M-H]^−^	C_15_H_10_O_5_	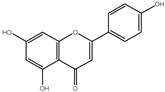 apigenin	[44,45,55]
9.49	809	[M-H]^−^	C_42_H_66_O_15_	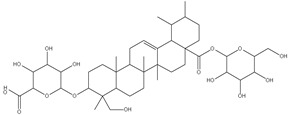 cynarasaponin E	[45]
11.30	307	[M-H]^−^	C_18_H_28_O_4_	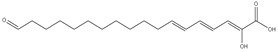 hydroxyoxo-octadecatrienoic acid	[45]
**Stem**
1.01	353	[M-H]^−^	C_16_H_18_O_9_	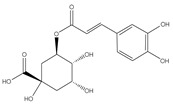 chlorogenic acid	[56]
6.73	593	[M-H]^−^	C_27_H_30_O_15_	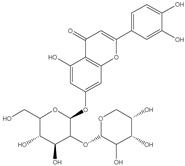 luteolin- 7-*O*-neohesperidoside	[56]
6.73	593	[M-H]^−^	C_27_H_30_O_15_	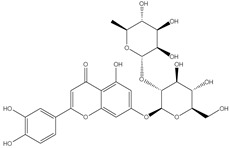 luteolin-7-*O*-rutinoside	[56]
7.06	461	[M-H]^−^	C_21_H_18_O_12_	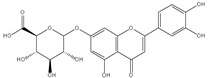 luteolin-7-*O*-glucuronide	[57]
7.41	515	[M-H]^−^	C_25_H_24_O_12_	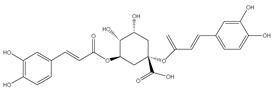 cynarin	[56,57]
8.18	779	[M-H]^−^	C_41_H_64_O_14_	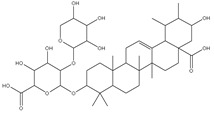 cynarasaponin F	[52]
8.36	269	[M-H]^−^	C_15_H_10_O_5_	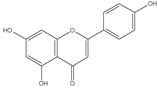 apigenin	[56]

## Data Availability

Data is contained within the article.

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
