# Peer review of "Comparative Estimation of the Cytotoxic Activity of Different Parts of Cynara scolymus L.: Crude Extracts versus Green Synthesized Silver Nanoparticles with Apoptotic Investigation"

_pharmaceutics, 2022, doi:10.3390/pharmaceutics14102185_

Round 1

Reviewer 1 Report

In this paper, the crude extracts from stem, flower and bract of Cynara scolymus L. were prepared, and the green synthesis of three kinds of silver nanoparticles was realized based on these three crude extracts. MTT assay was used to compare the cytotoxic activity of the six group. The results showed that AgNPs prepared from flower extract had more potent cytotoxicity against PC-3 and A549 cell lines than others, and its mechanism for apoptosis-induction was further investigated by Annexin V/PI staining. Further, RT-PCR and Western blotting were used to confirm apoptosis-induction of AgNPs of flower extract on A549 cells at the gene and protein expression levels, respectively. Finally, the bioactive metabolites in the crude extracts of flowers, bracts and stems were identified using UPLC-ESI-MS. The motivation behind the problem investigated in this manuscript is meaningful, and the experimental proof process is complete. However, there are some problems in the manuscript that need to be revised. The detailed comments are as follows.

(1)The introduction part of this paper gives a relatively comprehensive introduction to Cynara scolymus L., but the introduction of the green synthesis of AgNPs is not complete enough. The principle of synthesizing AgNPs from crude extract requires a clearer explanation. At least the reduction of silver ions by bioactive substances in plants should be included. This process is briefly mentioned in section 3.2 as an explanation of the UV spectral results, but I suggest more detail could be added in the introduction section.

(2)The characterization results of AgNPs are not fully presented. Three kinds of AgNPs were prepared in this paper, but only one of them was shown in the UV-visible spectrum and TEM images. And the description in the figure note was not clear enough. What does "YL" in Figure 1 and "LL" in Figure 2 stand for? It would be better to replace them with specific names of the different parts of the Cynara scolymus L. If alphabetical abbreviations are required, specify them in the manuscript.

(3)The key words in the manuscript should be consistent, at least without ambiguity. The descriptions of “AgNPs-flower””AgNPs-formula of flower extract”, “nano-flower””nano-flower extract” appear successively in the text part and the picture part of the manuscript. After reading the whole article, I know they represent the flower extract AgNPs, but in reality, the meaning behind these words are quite different.

(4)The quality of Cynara scolymus L. extracts obtained from different habitats and different extraction methods may be different, which should be fully studied.

(5)Standard synthetic AgNPs should be used as the control group in all the studies to better illustrate the effect of green synthesized AgNPs.

(6)Figure 6. (A) is incomplete and lacks the necessary annotations. The gel image should be appended with the name of the corresponding protein. Moreover, the band of the reference protein β-actin should also be displayed.

(7)A small part of the manuscript is missing. In section 2.5.2, it was mentioned that cell cycle analysis would be carried out, but no relevant content was found in the result section, which only described the apoptosis of cells.

(8)The format of the figures is not uniform. Before submitting the manuscript be sure that your material is properly prepared and formatted. Please check the size of the pictures in the manuscript.

(9)The identification of bioactive metabolites may not be necessary, or more relevant studies are needed to prove its significance.

(10)There are some small mistakes in this manuscript and need more corrections. For example, "AgNO3" in section 2.3, "CaCl2" and "IC50" in section 2.5.2. More attention should be paid to the writing of subscripts.

(11)The expression in this article is too vague to be easily understood and should be corrected in a more accurate description.

Reviewer 2 Report

The manuscript "Comparative Estimation of the Cytotoxic Activity of Different Parts of Cynara scolymus L.: Crude Extracts versus Green Synthesized Silver Nanoparticles with Apoptotic Investigation" is devoted to characterization of extracts of Cynara scolymus L. Total phenolics content, LC-ESI-MS analysis, and various methods for biological activity estimation were used. Anticancer potential of the leaf extracts was obtained. The results are interesting for pharmaceutics and plant chemistry.

The manuscript is written clearly, well-structured and has a good scientific soundness.

I think, this manuscript can be published in the Pharmaceutics after minor revision taking into account and some of the remarks described below:

1.     Add, please, the reagents list with their purity and manufacturer.

2.     It would be better to explain the Conclusion.

Round 2

Reviewer 1 Report

The authors have fully addressed my concerns. I think it can be accepted now.